# Fasting Mimicking Diet for Metabolic Syndrome: A Narrative Review of Human Studies

**DOI:** 10.3390/metabo15030150

**Published:** 2025-02-23

**Authors:** Alina Delia Popa, Andreea Gherasim, Laura Mihalache, Lidia Iuliana Arhire, Mariana Graur, Otilia Niță

**Affiliations:** 1Internal Medicine II Department, Faculty of Medicine, University of Medicine and Pharmacy “Grigore T Popa”, 700115 Iasi, Romania; alina.popa@umfiasi.ro (A.D.P.); laura.mihalache@umfiasi.ro (L.M.); lidia.graur@umfiasi.ro (L.I.A.); otilia.nita@umfiasi.ro (O.N.); 2Faculty of Medicine and Biological Sciences, University “Ștefan cel Mare” of Suceava, 720229 Suceava, Romania; graur.mariana@gmail.com

**Keywords:** metabolic syndrome, fasting-mimicking diet, obesity, type 2 diabetes, dyslipidemia

## Abstract

Metabolic syndrome (MetS) is an association of risk factors that share insulin resistance (IR), exerting a super cumulative effect on the risk of developing cardiometabolic diseases. Lifestyle optimization is a key element in the prevention and non-pharmacological therapy of MetS. Certain studies have concluded that some dietary patterns could be more beneficial as an adjunctive treatment for MetS. Fasting mimicking diet (FMD) is a form of periodic fasting in which caloric intake is restricted for 5 days each month. It has been studied for its beneficial effects not only in patients with neoplasia and neurodegenerative diseases but also for its effects on IR and metabolism. In this narrative review, the effects of FMD in patients with MetS were analyzed, focusing on its impact on key metabolic components and summarizing findings from human studies. FMD has demonstrated beneficial effects on MetS by reducing BMI and waist circumference, preserving lean mass, and improving the metabolic profile. Moreover, individuals with a higher BMI or a greater number of MetS components appear to derive greater benefits from this intervention. However, limitations such as high dropout rates, small sample sizes, and methodological constraints restrict the generalizability of current findings. Further large-scale studies are needed to confirm these effects and establish FMD as a viable non-pharmacological strategy for managing MetS.

## 1. Introduction

Metabolic syndrome (MetS), first described by Raven in 1988, is an association of cardiovascular risk factors that share insulin resistance (IR) [1], exerting a super cumulative effect on the risk of developing diabetes and cardiovascular or cerebrovascular disease [2,3]. The diagnosis of MetS has undergone revisions over time in terms of the number and type of inclusion criteria. Thus, in 2009, Alberti et al. [4] considered the presence of any three of the following changes to be necessary for diagnosing this condition: abdominal obesity (waist circumference (WC) ≥ 94 cm in men and ≥80 cm in women), hypertriglyceridemia (serum triglycerides ≥ 150 mg/dL or administration of specific medication to reduce triglycerides), low HDL cholesterol (<40 mg/dL in men or <50 mg/dL in women, or administration of specific medication), hypertension (blood pressure ≥ 130/85 mmHg or history of antihypertensive treatment), hyperglycemia (fasting plasma glucose (FPG) ≥ 100 mg/dL or pre-existing diagnosis of type 2 diabetes (T2DM). Although the National Cholesterol Education Program—Adult Treatment Panel III (NCEP-ATP III) [5] recommends different cutoff values for WC, the International Diabetes Federation criteria for MetS are better predictors of T2DM and cardiovascular disease. Beyond the disputes related to the genetic, epigenetic, behavioral, and environmental triggers of the pathogenic chain that lead to the appearance of MetS elements, visceral obesity is seen as a central catalyst that promotes chronic low-grade inflammation, IR, and neurohormonal changes [2,3]. All these abnormalities contribute to worsening MetS prognosis, amplifying adverse effects, and increasing cardiovascular risk [6].

A recent meta-analysis showed a wide range of variations in the global prevalence of MetS in the adult population (12.5–31.4%), which varied depending on the definition considered [7]. It is estimated that one-quarter of the European population is affected by MetS [8], while in the United States the prevalence increased from 37.6% in 2011–2012 to 41.8% in 2017–2018 [9].

Lifestyle optimization is a key element in the prevention and non-pharmacological therapy of MetS [8,10]. The Fasting Mimicking Diet (FMD) is a form of periodic fasting designed by Valter Longo, in which caloric intake is restricted for 5 days each month (1090 kcal on the first day and 725 kcal on the remaining 4 days). The foods recommended in FMD comprise plant-based ingredients, such as vegetable soups, seaweed oil, herbal tea, olives, cabbage biscuits, and energy or chocolate bars. It is a form of ketogenic diet in terms of macronutrient composition (on day 1 containing 10% protein, 56% fat, 34% carbohydrates, and on days 2 to 5–9% protein, 44% fat, 47% carbohydrates) [11].

A recently published bibliometric review of the literature identified 169 articles on FMD, noting a constant and significant increase in the number of articles published from 2017 to 2023, thus suggesting a growing recognition of the relevance and importance of FMD in the scientific community. Cluster analysis of the keywords showed that most articles studied the effects of FMD on oncology (26.64% of publications). However, a significant proportion of the articles were dedicated to nutrition and metabolism (18.34%) [12].

These results motivate the evaluation of the metabolic effects of FMD in patients with MetS and the comparison of this nutritional therapy with other lifestyle optimization interventions for MetS [13,14]. For example, FMD-induced changes in cardiovascular risk factors are comparable to those produced by the Mediterranean diet [15]. Direct comparisons are needed to determine whether FMD offers advantages over existing approaches in terms of efficacy, safety, and tolerance for developing clinical practice guidelines and decision-making for individualized patient care [12,14].

We conducted a narrative review to comprehensively examine the effects of FMD in patients with MetS as well as on the main components associated with this entity: obesity, IR, hyperglycemia and T2DM, dyslipidemia, high blood pressure, metabolic associated fat liver disease (MAFLD), and gut microbiota changes. Our focus was on the results provided by human studies to establish the current knowledge and gaps related to the effects of FMD on metabolic health and its potential therapeutic role in clinical practice.

## 2. Material and Methods

We performed a systematic search in two databases, PubMed and EBSCO, using the term “fasting-mimicking diet” and a combination of keywords: “metabolic syndrome”, “obesity”, “hyperglycemia”, “insulin resistance”, ”dyslipidemia”, “high blood pressure”, “gut microbiota”, “metabolic associated fat liver disease”, and “diabetes”. Using this search strategy, we identified 193 studies. After excluding the duplicates (*n* = 47), screening for titles and abstracts (*n* = 72), and full texts (*n* = 58), we selected 16 studies addressing FMD in humans with MetS or its components. We opted for a narrative review due to the heterogeneity of the studies’ design, length, and number of participants.

## 3. Metabolic Adaptations During Fasting

In the postabsorptive period, glucose is used by tissues for energy or stored in the liver as glycogen [16]. Glycogen genesis is under the influence of the circadian rhythm, with GSK3 (Glycogen synthase kinase 3) activity modulated by transcription factors located in the suprachiasmatic nucleus (SCN) of the anterior hypothalamus: Circadian Locomotor Output Cycles Kaput (CLOCK) and Brain and Muscle ARNT-Like 1 (BMAL1) [17]. The decrease of blood glucose levels below 80 mg/dL inhibits insulin secretion and triggers glucagon release, stimulating hepatic glycogenolysis to maintain normoglycemia [17].

Secor et al. [18] described three distinct phases during prolonged fasting corresponding to different sources of energy, each characterized by different metabolic mechanisms. During the transition from the post-absorptive to the fasting state, the energy metabolism switches from glucose (produced initially by glycogenolysis and afterward to gluconeogenesis) to ketones derived from fatty acids [19]. In the initial phase, which lasts several hours after the start of fasting, a decrease in metabolic rate and depletion of hepatic glycogen stores are observed. This leads to the use of alternative energy sources [20] representing a transition from glucose use to lipid oxidation as the main source of energy [21,22]. This transition is marked by an increase in the concentration of circulating ketone bodies, which serve as an alternative fuel for the brain and other tissues [23]. The duration of fasting significantly influences the extent of these adaptations; prolonged fasting causes a more pronounced transition to ketogenesis and a greater dependence on fat stores as an energy source [24]. After approximately 60 h of fasting, a process of re-esterification of free fatty acids in adipose tissue occurs, causing 40% of them to be recycled as triglycerides and stored. The remaining 60% are released into the blood to be oxidized in other tissues, such as muscle and liver [19]. In this context, lipid utilization becomes a significant energy source, preserving amino acid reserves. Ketone bodies also play a crucial role in providing energy for nervous and cardiac tissues, which normally prefer glucose as the main energy source [18]. The critical roles played by the mitochondrial tricarboxylic acids cycle (TCA) and hepatic lipogenesis require increased levels of Branched-Chain Amino Acids (BCAAs), which are produced mainly in muscle. Increased concentrations of butyrate, BCAAs, and acylcarnitine in the blood are recognized as indicators of fasting [25].

The third phase of fasting is a critical stage in which the transition from lipid to protein catabolism occurs, the production of ketone bodies decreases, and gluconeogenesis from amino acids increases [18]. Prolonged fasting (>24 h) is associated with a marked decrease in peripheral insulin sensitivity [23] and a reduction in insulin secretion relative to insulin sensitivity (insulin secretion disposition index) [26]. The decrease in insulin secretion is a beneficial adaptive mechanism known as the “beta cell resting” phenomenon [26,27]. Although the decrease in insulin sensitivity may seem like an undesirable effect, the mechanism of IR during prolonged fasting differs from that observed in T2DM [27], being related to a reduction in hepatic glucose output [26,28] and representing an adaptive response to a lack of energy intake [29]. During prolonged fasting, the decrease in circulating insulin and C-peptide levels is mainly due to improved hepatic insulin action [26]. An increase in growth hormone has been reported during this phase, which decreases with the resumption of food intake and is thought to have long-term beneficial effects on health [30].

Furthermore, fasting induces changes in the expression of various genes involved in metabolic regulation [31]. These changes impact pathways related to nutrient sensing (e.g., mTOR—the mechanistic (or mammalian) target of rapamycin, AMPK—adenosine monophosphate-activated protein kinase, SIRT1—sirtuin 1 protein coding gene) [32]. The activation of AMPK, a key regulator of energy homeostasis, is thought to play a crucial role in promoting metabolic adaptations during fasting [31]. In the skeletal muscle, prolonged fasting leads to decreased insulin sensitivity, initially attributed to increased free fatty acids levels and substrate competition [33,34]. However, research suggests that mitochondrial dysfunction may be a secondary consequence of IR rather than a primary cause [33]. The impact of fasting on the liver includes changes in hepatic lipid metabolism, leading to increased and decreased levels in certain lipid species depending on the duration of the fast [21,35]. The thyroid axis also undergoes modifications, with decreased serum T3 and T4 levels in some studies, while TSH levels remain relatively stable or even increase [36,37]. These changes are believed to be mediated by the central melanocortin system and leptin, a hormone involved in energy balance [36,38,39]. However, the precise regulatory mechanisms remain incompletely understood, particularly regarding the tissue-specific differences in thyroid hormone metabolism [39].

FMD has been studied for its beneficial effects not only in patients with neoplasia and neurodegenerative diseases but also for its effects on IR and metabolism. Thus, FMD inhibits cell growth by regulating the AMPK-mTOR signaling pathway [13] and induces the phosphorylation of AMPK, a fundamental enzyme involved in mitochondrial biogenesis [13,40]. Ketogenic diets have been studied in oncology starting from the observation that tumors cannot use ketone bodies because they do not express β-hydroxybutyrate dehydrogenase or succinyl-CoA:3-ketoacid CoA transferase [32,41], thus being deprived of the energy they need for progression and survival. In animal studies, FMD modulates the expression of neurogenin 3 (Ngn3), an early developmental marker involved in the regeneration of insulin-producing β-cells [13,42,43], improves neuronal survival and differentiation by inducing the expression of NeuroD1 [13], stimulates oligodendrocyte regeneration and differentiation [44], and protects Oligodendrocyte Precursor Cells from apoptosis [13].

## 4. Fasting-Mimicking Diet and Its Influence on Metabolic Syndrome Components

### 4.1. Impact of Fasting-Mimicking Diet on Obesity

The diagnosis and evaluation of obesity have changed in recent periods. It is considered that the body mass index (BMI), initially used as diagnosis criteria, is not enough indicative for its echo on general health. WC is a better estimator of metabolic effects, being closely related to IR and thus with its comorbidities, such as MAFLD, dyslipidemia, or T2DM [45,46]. According to Canadian Adult Obesity Clinical Practice Guidelines, in 2020 obesity was considered “a complex chronic disease in which abnormal or excess body fat (adiposity) impairs health, increases the risk of long-term medical complications and reduces lifespan” [47].

Most studies on FMD showed a positive effect on weight and BMI. In participants with normal weight, the mean BMI changed after three FMD cycles across three months; the mean weight loss was 3% in a pilot study [11] and 2.6 kg in two randomized control trials (RCT) [48,49], returning to the baseline value after refeeding [49]. A similar effect was observed in metabolically healthy overweight participants [50], with a weight loss of 2.8 ± 0.2 kg and subsequent reduction in BMI compared to baseline. Although weight loss was observed in all BMI categories, the extent of this change depended on the weight status at the beginning of the study, with participants with a BMI > 30 kg/m^2^ experiencing the biggest change [48]. In patients with obesity and T2DM, BMI was significantly reduced in the FMD group compared to the control group [51,52,53,54]. WC reduced significantly in the FMD group (from 105.10 ± 5.20 cm to 90.82 ± 4.26 cm) compared to the control group (from 104.32 ± 4.84 cm to 98.38 ± 4.27 cm) [51]. The same significant trend was noticed when combining FMD with the DASH diet [55], but not in the DASH alone group. BMI and weight showed an important drop after 6 months of FMD in patients with T2DM and increased albumin-to-creatinine ratio; also, the participants allocated to a Mediterranean diet did not have a significant weight or BMI change [56]. However, this positive effect on anthropometric indices was not noticed in all studies [55]. Although Sadeghian et al. did not report a significant weight change (CER −2.29 ([SD], 1.95) kg vs. −1.13 (2.27) kg for FMD (*p* = 0.06), that may indicate a negative trend. However, the basal metabolic rate (BMR) registered a more relevant drop in the caloric energy restriction (CER) group in the same study [57] (Table 1). 

Weight loss is a common consequence of FMD, but the proportion of fat mass reduction has varied across studies. While some studies showed the preservation of lean mass, others indicated its loss, raising concerns about potential adverse effects. In some studies, a reduction in fat mass and preservation of lean mass in normal-weight subjects was observed [48,49,60]. However, other studies did not report significant changes in visceral fat among normal-weight subjects [58]. Obese individuals have shown decreases in fat mass and visceral fat, with the preservation of lean mass in most studies [59]. However, some interventions have been associated with modest losses in lean mass, especially in the context of very restrictive diets [58]. In patients with T2DM, FMD has been effective in reducing fat mass [53], but a modest decrease in lean mass was registered by Van der Velden et al. [61]. After a 3-month follow-up period, a small study using MRI in 15 participants showed a decrease in subcutaneous adipose tissue SAT (*p* = 0.008) and visceral adipose tissue VAT (*p* = 0.002), with a more pronounced decrease of VAT in overweight participants (*n* = 10, *p* = 0.003) [60]. Also, some studies have not identified relevant changes in VAT in this group [56]. A similar result was observed at the end of a 3-month follow-up period when participants on a Mediterranean diet (MD) lost a significant amount of fat-free mass (FFM) (MD: −4.7 ± 1.3 lbs, *p* = 0.0008), an effect which was not observed in the FMD group (FMD: −0.9 ± 1.5 lbs, *p* = 0.5466); FMD vs. MD, *p* = 0.0498) [59]. These data suggest that tailored diets can significantly contribute to improving metabolic health, although some negative or inconsistent results highlight the need for further research to optimize protocols to fully understand the long-term impact of FMD on body composition and to minimize lean mass loss [23] (Table 1).

Changes in body composition and adipose tissue distribution are relevant for assessing the metabolic risk modification resulting from dietary therapy. Adipose tissue is regarded as an immune organ because it contains local inflammatory cells, including macrophages, CD4+ Th2 T cells [62], and eosinophils, which participate in the onset of local inflammation and subclinical chronic inflammation that contribute to metabolic disorders, elevated free fatty acids, and alterations in adipokine secretion [63]. Furthermore, a weight-loss-centric approach is seen as only superficial and prone to failure. The model fails to consider the intricate nature of adipose tissue functions, which extend beyond just a mere energy storage site. Beyond the body weight changes, FMD has been shown to affect other relevant metabolic markers. For instance, Brandhorst et al. [60] reported that three cycles of FMD were associated with a decrease in biological age, independent of weight loss, based on a validated predictive measure of morbidity and mortality. These effects were linked to reductions in IR, hepatic fat, and improvements in the lymphoid-to-myeloid ratio, an indicator of immune system age [60]. Other studies have observed changes in levels of various hormones and growth factors, such as insulin-like growth factor 1 (IGF-1) [50] and leptin [15], following FMD interventions. The study conducted by Sadeghian et al. [57] showed an increased leptin level after FMD, but not at the end of CER. Although not significant, the increased level of leptin following FMD after weight loss may represent a strong tendency for weight regain [57]. The reduction of IGF-1 was more pronounced in participants with higher baseline IGF-1 levels (≥225 ng/mL), and the decrease was significantly larger compared to those with lower baseline levels. The authors’ explanation was based on the long-lasting effects of the low protein/amino acid content of the FMD [48] (Appendix A—Table A1).

Canadian Adult Obesity Clinical Practice Guidelines recommend different medical nutrition therapies for obesity [64], such as medical nutrition therapy by a registered dietitian, intensive behavioral therapy, and dietary patterns or food-based approaches. Although being associated with higher attrition rates and more adverse events, intermittent fasting was included in the proposed medical nutrition therapies, showing comparable outcomes in terms of weight, fat mass, fat-free mass, WC, glucose, HbA1C, triglycerides, and HDLc changes [65]. The Edmonton Obesity Staging System (EOSS), which considers metabolic, physical, and psychological factors, has been suggested as a five-stage classification system for obesity. Its purpose is to assist healthcare professionals in making informed decisions regarding obesity therapy [47]. MetS includes not only central obesity but also other metabolic conditions, corresponding to stage 2 or more according to the EOSS, indicating the need to associate pharmacological therapy with medical nutrition therapy. In 2021, Acosta and colleagues [66] introduced a novel approach to categorizing obesity that emphasizes the underlying mechanisms responsible for weight increase, as opposed to the conventional methodologies that rely on body measurements and metabolic traits [3]. They identified four distinct obesity phenotypes based on specific measurement thresholds: abnormal satiation (“hungry brain”), abnormal hedonic eating or emotional hunger, abnormal satiety (“hungry gut”), and low predicted energy expenditure (“slow burn”) [66]. According to this study, current medical therapies are not effective for all patients. Individualized lifestyle and pharmacological interventions customized to each phenotype are recommended to provide a personalized approach to managing obesity [67]. Most studies have concluded that FMD is associated with weight loss and preserved lean mass, indicating it is a potential adjunctive therapy for Mets. However, there is a lack of evidence of the long-term effects of FMD on weight regain, and studies with a prolonged follow-up period are needed.

### 4.2. Effects of FMD on Insulin Resistance and Diabetes

FMD may benefit patients with chronic metabolic-related diseases such as T2DM and MAFLD [12]. Cheng et al. [43] found that a 4-day FMD in mice has been shown to stimulate the regeneration of β cells, suggesting its potential contribution to the improvement in both type 1 and type 2 diabetes. The author also performed a study on a small sample of human subjects and on human pancreatic β-cells, which highlight that the inhibition of mTOR, along with PKA, promotes Ngn3-driven β-cell regeneration in human type 1 diabetes islets [43]. Under fasting conditions PKA and mTOR activity in human pancreatic islets with type 1 diabetes decreases, stimulating the expression of specific genes and subsequent insulin production [68]. mTORC1 plays an essential role in the survival and function of human pancreatic β cells by modulating autophagy and reducing endoplasmic reticulum stress [69]. Fasting influences mTORC1 and mTORC2, affecting glucagon secretion and energy homeostasis in mice with tissue-specific deletion of the mTORC1 regulator Raptor in α cells (αRaptorKO) [70]. Multiple other studies performed on animal models underscored the potential of fasting in influencing endocrine pancreatic function. The inhibition of mTOR promotes autophagy by recycling cellular components and reducing oxidative stress, facilitating the adaptation of pancreatic β-cells to the conditions of reduced energetic intake [71]. This process contributes to maintaining optimal insulin secretion and protecting against endoplasmic reticulum stress-induced apoptosis, highlighting the essential role of mTOR in regulating energy homeostasis and preventing the progression of metabolic dysfunction in diabetes [72]. The interaction between the mTOR signaling pathway and the PKA pathway is achieved through the phosphorylation of the Raptor protein, a key component of the mTORC1 complex [73]. This phosphorylation is essential for glucose metabolism, with mice with a phosphorylation-resistant Raptor mutation exhibiting metabolic dysfunction, including IR [73]. On the other hand, PKA can inhibit mTORC1 activity through the same phosphorylation site, thus suggesting a dual regulation depending on the cellular context [74]. The mTORC1 complex acts as a metabolic rheostat, integrating signals from nutrients, energy, and growth factors to regulate processes such as protein synthesis and autophagy, thereby influencing cell growth and metabolism [75].

In normal-weight non-diabetic participants, the decrease in FPG after a cycle of FMD was accompanied by improved insulin sensitivity and a reduced homeostatic model assessment to measure insulin resistance (HOMA-IR) index [50]. After 3 months of FMD, FPG levels decreased by 11.3% ± 2.3% (*p* < 0.001) and remained lower than baseline levels after completing the third FMD cycle [11]. Although this effect was not consistently found [48,76], Wei et al. showed reduced FPG in individuals with metabolic risk [48]. However, a significant reduction in serum FPG levels was observed in subjects in the CER group but not in those following FMD (*p* < 0.001), in an RCT involving metabolically healthy women with obesity. No statistically significant within-group changes were observed in serum insulin levels, HOMA-IR, or QUICKI in either the FMD or CER group [57]. Although the inconsistent effects on FPG, results from a small sample of patients with prediabetes included in an RCT were encouraging, FMD conducting to a decrease of HOMA-IR (*p* = 0.046), and HbA1c levels (*p* = 0.032) [60]. A similar effect was observed in an RCT with a larger sample of patients with MetS [59]. Patients on FMD had a decrease in HOMA-IR (*p* = 0.0475) and serum glucose levels (*p* = 0.0488), but not in the HbA1c level (*p* = 0.0059). During the follow-up period, FMD decreased insulin (−5.6 uU/mL; *p* = 0.0046), HbA1c (−0.1; *p* = 0.0116), and HOMA-IR (−1.5; *p* = 0.0066) levels as compared to baseline [59] (Table 2).

In patients with diabetes, FMD consistently improved glucose metabolism, insulin sensitivity, and long-term glycemic control, with notable reductions in HbA1c and fasting glucose. Most studies [51,52,53,61] showed that FMD significantly reduced HbA1c levels. A study using the Mediterranean diet as a control group showed a significant reduction of HbA1c in patients on FMD (8.0 ± 0.4% vs. 6.7 ± 0.3%; *p* < 0.001) [52] (Table 2).

Several studies [53,56] reported significant reductions in HOMA-IR and FPG in participants following FMD compared to control groups. The improvements were sustained over multiple diet cycles, with a reduction in the need for antihyperglycemic medication. Sulaj et al. [56] showed that antidiabetic medication could be decreased in 57% of participants in the FMD group after three diet cycles compared to 32% of the Mediterranean Diet group. Furthermore, glucose-lowering therapy could be reduced in 67% of participants in the FMD group compared to baseline after six diet cycles, whereas in the Mediterranean Diet group, the dose of antidiabetic agents had to be increased in 21% of the participants [56]. Similar findings were obtained in a more recent RCT [53]. Although the dose of antidiabetic drugs was reduced in 40% of participants in the FMD group, it increased in 44% (*n* = 17) of control group participants (*p* < 0.001) after 12 months. Moreover, anti-diabetic medication was completely stopped in 16% (*n* = 7) of the participants in the FMD group, while additional medication was necessary in 26% of patients (*n* = 10) of the control group (*p* = 0.006; Table 2).

Currently, many fasting interventions are available, and some studies have compared their effect on different parameters [23,77,78]. Among these studies, one meta-analysis conducted by Wen et al. [78] focused on comparing the effects of different types of intermittent fasting (twice-per-week fasting, FMD, time-restricted eating, and periodic fasting) on IR and glycemic control in patients with T2DM. All intermittent fasting interventions demonstrated superiority to the conventional diet, with no difference between them. The superiority was attributed to various factors, including weight loss, changes in lipid profiles, and modulation of hormonal pathways [78].

These findings suggest that FMD may affect metabolic health beyond weight loss. However, the relatively small sample sizes in some studies [52,60] and the variations in FMD design (proof of concept, pilot studies, RCT) limit the ability to draw definitive conclusions. At the same time, the research’s protocols knew a wide variation. Although most studies included control groups, these were diverse, including caloric energy restriction [57], standard diabetes diet [61], DASH [55], Mediterranean diet [52,56,59], or standard meal replacement [51]. Moreover, it is important to consider the limitations and uncertainties surrounding using FMD in T2DM management. Not all studies had a follow-up period [51,52,53,54], and usually, this was limited to 3 months [56] or 6 months [61]. Long-term studies are needed to assess the sustainability of the observed benefits and to evaluate potential long-term risks or adverse effects. The potential for individual variability in response to FMD is also a critical factor that requires further investigation [11]. Some studies have shown inconsistent findings, with improvements in some metabolic parameters but not others [13]. This highlights the need for personalized approaches to FMD interventions, considering individual metabolic characteristics and health status.

### 4.3. Atherogenic Dislipidemia

In patients with obesity and hypertrophic and dysfunctional adipose tissue, free fatty acids inhibit insulin receptor signaling pathways and enhance IR. This effect is triggered by the excessive secretion of proinflammatory adipokines such as leptin, resistin, and TNF-α [79]. This high level of free fatty acids originates from increased lipolysis in the adipose tissue as a consequence of IR [80]. In the liver, this high amount of free fatty acids is metabolized through β oxidation or is used to synthesize triglycerides, which are incorporated into very low-density lipoproteins (VLDL). When these pathways are exceeded, hepatic steatosis develops through the accumulation of triglycerides in the hepatocytes. Higher plasma levels of free fatty acids correlate with the overproduction of VLDL. The clearance of the VLDL particles is inhibited, leading to hypertriglyceridemia and a high level of small and dense LDL particles [81]. All these metabolic changes facilitate the development of hyperglycemia and hyperinsulinemia, which are key players in the pathophysiology of dyslipidemia [79].

In normal-weight participants, Videja et al. [50] found that both FMD and a 5-day regular diet supplemented with four servings of vegetables (VEG) improved lipid profile. These results were similar when comparing FMD with a normal diet in metabolic healthy participants with mixed BMI [48]. However, when controlling for the drop-out, FMD showed a more significant increase in HDLc levels (*p* = 0.03) [48].

In patients with T2DM FMD was not associated with significant changes in LDLc, HDLc, or triglycerides when compared to a Mediterranean diet [52,56]. Similar results were obtained in a study comparing 2-month FMD cycles (five days per month) with usual care for T2DM [53]. The notable exception was for an adjusted estimated treatment effect on HDLc, which increased with 0.1 mmol/L (95% CI 0.0–0.2, *p* < 0.001). Furthermore, the use of cholesterol-lowering medication remained stable over 12 months in most participants (80% of participants receiving FMD vs. 84% of control participants) [53]. However, when compared to standard meal replacements, FMD increased HDLc and reduced triglycerides, LDLc, and total cholesterol, showing significant statistical intergroup differences (all *p* < 0.05) [51] (Table 3).

In patients with MetS, both FMD and the Mediterranean diet improved total cholesterol levels, but only the Mediterranean diet improved HDLc level (*p* = 0.0418) [59]. A modified fasting therapy (MFT) using fermented medicinal herbs and exercise (400–600 kcal/day) [58] decreased total cholesterol from 181.5 ± 17.4 to 156.0 ± 12.7 mg/dL and triglycerides from 103.9 ± 22.8 to 90.5 ± 18.4 mg/dL (Table 3).

Studies with longer duration [53] showed some positive effects on HDLc, but no major changes in other lipids. Shorter studies [50] reported immediate benefits only on triglycerides, with no significant impact on other components of the lipid profile. FMD has the potential to improve the lipid profile, particularly by reducing total cholesterol and triglycerides, being more effective in obese and prediabetic populations. The effects on HDLc and LDLc were variable, being influenced by diet composition, duration of the intervention, and the characteristics of the population. Not all patients responded favorably, and in some cases (patients with T2DM with nephropathy [56]), the effects may be negligible.

### 4.4. Effects of FMD on Blood Pressure

Hypertension has a substantial contribution to cardiovascular disease and mortality [82]. Lifestyle modifications, including dietary changes, are essential interventions in the management of hypertension [82]. FMD has emerged as a potential non-pharmacological approach to improving various health markers, including blood pressure (BP) [83]. It induces a state of caloric restriction that can lead to weight loss and subsequent reduction of BP [84], but may also influence BP independently of weight loss [82] through the modulation of the renin-angiotensin system [85]. Intermittent fasting may affect this system, potentially leading to decreased angiotensin II levels and increased activity of angiotensin-converting enzyme 2 [86]. Another potential mechanism involves the reduction of inflammation associated with hypertension [86], as some studies showed a decrease in C-reactive protein (CRP) levels in participants on FMD [48,60].

In generally healthy adults, FMD reduced systolic (4.5 ± 6.0 mmHg; *p* = 0.023) and diastolic BP (3.1 ± 4.7 mmHg; *p* = 0.053 between groups) as compared to the control group on a normal diet [48]. In this study inflammatory marker CRP (*p* = 0.27) did not differ significantly between groups [48] (Table 4).

In patients with MetS, FMD reduced systolic BP (*p* < 0.0001) and CRP (*p* < 0.0001) [62]. Another study on patients with MetS showed that fasting followed by a modified DASH led to a sustained reduction both in 24-h ambulatory systolic BP and mean arterial pressure (MAP) (*p* < 0.05) [55]. Moreover, 43% of subjects undergoing fasting were able to significantly reduce (*p* = 0.035) their antihypertensive medication intake while maintaining controlled blood pressure, compared to only 17% of those following the DASH diet alone [55]. In T2DM patients, there were inconclusive results. In a recent study comparing 12-month FMD cycles with usual care, there were no treatment effects on systolic BP (−0.4 mmHg; 95% CI −6.0, +5.1; *p* = 0.87), diastolic BP (−0.5 mmHg; 95% CI −3.2, +2.2; *p* = 0.74), or in antihypertensive medication after 12 months [53]. In another study, systolic BP and diastolic BP significantly decreased after FMD, with results not observed in the control group on standard meal replacements [51] (Table 4). In T2DM patients with nephropathy, antihypertensive medication was reduced by 10% in the FMD group compared to 5% in the Mediterranean diet group, after three and six diet cycles. FMD led to a reduction in albumin to creatinine ratio in patients with microalbuminuria levels at baseline (*p* ≤ 0.05), but not in those with macroalbuminuria (*p* = 0.23). Furthermore, sUPAR (soluble urokinase-type plasminogen activator receptor), a marker of persistent systemic inflammation, decreased by 156 pg/mL after six cycles in the FMD group [56]. Recent studies suggest a correlation between elevated sUPAR levels and MetS, obesity, and IR, indicating a potential role in inflammation-related metabolic dysfunction [87,88].

Despite the promising results of certain studies, a definitive conclusion regarding the beneficial effect of FMDs on blood pressure cannot be drawn. Sample size, length, and lack of follow-up periods represent the main limitations of the studies, making it difficult to accurately assess the effects of FMDs. Furthermore, the heterogeneity in control groups and participant characteristics makes this task even more challenging. Thus, the use of more standardized protocols regarding sample size, study duration, control groups, and follow-up would be crucial for improving the reliability and generalizability of future research. Another important aspect would be the evaluation of FMD efficacy and safety in diverse populations of patients with hypertension and comorbidities. Moreover, a mechanistic approach to the effects of FMD on blood pressure and its pathophysiology, focusing on the RAS, inflammation, or gut microbiome, could help elucidate mechanisms beyond weight changes and support the development of more individualized therapeutic approaches.

### 4.5. Metabolic-Associated Fatty Liver Disease (MAFLD)

Although not included in MetS, MAFLD is a prevalent chronic liver condition associated with insulin resistance and obesity, characterized by hepatic fat accumulation, inflammation, and fibrosis [89]. Its rising prevalence makes effective therapeutic interventions a stringent need. Despite evidence supporting pharmacological approaches, significant side effects limit the therapeutic choice. Thus, there is a need to explore non-pharmacological interventions, including FMD [89].

Animal studies have provided valuable insights into the potential benefits of intermittent fasting for MAFLD. A study investigating the effects of intermittent fasting in mice on a high-fat diet [90] showed improved lipid deposition and upregulation of the macrophage migration inhibitory factor (MIF)/AMPK pathway [90]. Although this suggested a potential beneficial effect of FMD in alleviating MAFLD-associated hepatic steatosis, current research on FMD on MAFLD remains limited.

In RCT in humans, FMD was associated with beneficial effects by improving the cytolysis syndrome, although it was associated with a non-significant decrease in transaminase levels in patients with MetS (ALT decreased from 30.1 ± 12.8 to 20.5 ± 4.2 IU/L; AST decreased from 24.1 ± 4.3 to 19.5 ± 2.9 IU/L (not significant) [58]. Nonetheless, the study’s limited sample size and absence of direct evaluation of liver markers restrict the conclusions that can be made about the precise impacts on MAFLD.

In a study conducted by Brandhorst that included healthy adults and patients with prediabetes, a reduction in hepatic fat fraction (HFF) (*p* = 0.049) was noticed in 15 participants. In five participants with steatosis, the percentage of liver fat registered a nearly 50% reduction, from 14.32 ± 5.8% to 7.94 ± 4.22% (*p* = 0.02) [60].

In healthy volunteers, FMD was associated with a significant decrease in trimethylamine N-oxide (TMAO) levels [50]. Elevated TMAO levels have been linked to an increased risk of MAFLD [50], and significantly higher TMAO levels have been detected in individuals with histologically confirmed MAFLD [91]. Despite the small sample size and the absence of a control group with similar caloric restriction, the positive effects on TMAO, along with improvements in other metabolic markers such as ketone bodies and IGF-1 [60], suggested that FMD may have beneficial effects on the metabolic factors involved in MAFLD development. Another relevant mechanism through which FMD may exert its effects on MAFLD is the modulation of gut microbiota composition and function [89,92]. Bile acid metabolism, lipid absorption, and inflammation, which are important factors in MAFLD pathogenesis, could be influenced by alterations in gut microbiota [92]. MetS induces significant alterations in the gut microbiome, leading to a distinct microbial composition associated with metabolic dysfunction. However, a recent RCT on patients with MetS showed that refeeding reverses these changes by promoting the growth of short-chain fatty acid (SCFA)-producing bacteria. However, fasting did not have a significant impact on gut microbiome alpha and beta diversity [55]. The beneficial effects of SCFAs on the gut-liver axes [93] are associated with the upregulation of glucagon-like peptide-1 (GLP-1) in enteroendocrine cells and the improvement of glucose tolerance [93]. In patients with MAFLD, downregulated GLP-1 receptor expression levels can be restored by the treatment with butyrate through the phosphorylation of hepatic AMP-activated protein kinase (AMPK) and the insulin receptor [93]. Liver macrophages, including Kupffer cells, monocyte-derived macrophages, and capsular macrophages, can undergo phenotypic changes in response to cytokines, fatty acids, endotoxins, and metabolites. These cells may adopt either a pro-inflammatory state (resembling classically activated M1-like macrophages) or an anti-inflammatory state (similar to alternatively activated M2-like macrophages). In the context of fatty liver disease, palmitic acid, a saturated fatty acid, promotes the polarization of macrophages towards the proinflammatory M1-like phenotype, which is reflected in increased production of TNF-α and IL-6 [94]. SCFAs exert beneficial effects on steatosis and liver inflammation by reducing the expression of tumor necrosis factor-α in a myeloid subset consisting of hepatic and Kupffer cells [93]. Additional research is needed to comprehend the relationship between FMDs and gut microbiota and its influence on the development and progression of MAFLD.

## 5. Discussions

FMD, as a medical nutrition therapy, showed beneficial effects on its components and on patients with MetS by reducing BMI and WC, preserving lean mass, and improving the metabolic profile. Furthermore, patients with higher BMI or with more components of MetS seem to benefit more from this intervention. However, FMD was associated with high dropout rates, ranging between 2.3% [50] and 30% [61]. This should be considered when opting for FMD as an alternative personalized diet for MetS. Brandhorst et al. [60] examined the effect of FMD on biological age, which was determined by combining multiple systemic biomarkers (albumin, alkaline phosphatase, serum creatinine, C-reactive protein, HbA1c, systolic blood pressure, and total cholesterol) into a single variable. Their findings indicate that FMD reduced median biological age by 2.5 years, independent of weight changes. Participants with higher metabolic risk, particularly those with elevated CRP levels, benefited the most from FMD, suggesting that individuals with increased inflammation may experience greater advantages from this intervention. However, this effect could potentially be attributed to regression to the mean and should be further investigated in larger studies. While FMD showed promising results in these patients, the sustainability of these diets over extended periods remained underexplored.

The number of cycles of fasting reported in the studies on the effect of FMD on MetS and its components has shown a broad variation, from 1 [50] to 12 cycles [53]. This variability limits the generality and potentially leads to heterogeneity of the results on weight, anthropometric indices, and metabolic markers. Glycemia or lipid profile can respond immediately after changing the diet composition, but the dietary effects of FMD on weight, HbA1c, and even inflammation markers need a longer period to be observed. Longitudinal studies with longer lengths are necessary to better understand the effect of FMD on the metabolic markers associated with MetS.

FMD is a short-term dietary intervention, which provides a low protein intake, with a high amount of healthy fats. At the same time, this diet brings the necessary complex carbohydrates, as well as essential vitamins and minerals to avoid potential side effects determined by essential nutrient deficits [95]. The low protein content of FMD aims to decrease the IGF-1 signaling pathway, which is an essential mechanism to promote longevity and cell resistance to stress. Previous studies have shown that high levels of IGF-1 are associated with an accelerated rate of aging and a high risk of cancer and metabolic diseases. The downregulation of the IGF-1/mTOR pathway enhances autophagy and cell-protection mechanisms [96]. Diets with a low content of proteins and specific amino acids, such as methionine, decrease IGF-1 signaling pathways and enhance mitochondrial function, leading to lifespan extension in animal models [97]. The high intake of unsaturated lipids provided by FMD is essential for ketogenesis triggering. Thus, the production of β-hydroxybutyrate and acetoacetate allows their use as an alternative fuel to glucose, mimicking the metabolic effects of prolonged fasting [98]. Not only that, the ketone bodies provide energy during caloric restriction, but at the same time, they exert neuroprotective and anti-inflammatory effects and also regulate gene expression by inhibiting histone deacetylases [99]. While strict fasting diets have a severe deficit of carbohydrates, FMD includes a controlled quantity of complex carbohydrates, which ensures a minimal glucose intake, necessary to maintain energy homeostasis without inducing a significant insulin rise [11]. Furthermore, this low level of carbohydrates helps reduce systemic inflammation and enhance insulin sensitivity, which are both important mechanisms in T2DM and MetS prevention [48].

Certain components of metabolic syndrome (MetS) present with gender-specific differences in prevalence, magnitude, and physiological mechanisms. These are determined by hormonal, genetic, and behavioral factors, and manifest differently in cardiovascular and metabolic risk in men and women. Men tend to accumulate more visceral fat [100], have higher triglyceride levels and lower HDL-C from a young age, leading to earlier cardiovascular events. In women, menopause marks a significant metabolic transition, with greater accumulation of visceral fat, insulin resistance, and hypertension. While LDL-C levels are a better predictor of cardiovascular disease in men [101], in women, risk is more closely correlated with triglycerides and inflammatory markers [102]. Another important limitation is the lack of comparisons of the efficiency of FMD according to gender, although these are important differences in the prevalence of metabolic syndrome components that can be differently addressed by FMD.

The main limitations of FMD studies in humans were related to the small sample size and short study duration, the representativity of the sample included in the analysis, methodological factors (such as a high dropout rate, limitations to certain patient categories, and gender imbalance), and the insufficient mechanistic analysis, which limits the generalizability of the results. These main gaps should be addressed by future research by conducting longitudinal studies to assess the long-term impacts of FMD on MetS and expanding research to include groups with diverse demographic characteristics and comorbidities. A deeper understanding of individual variations in response to FMDs could lead to the development of personalized approaches to MetS management.

## 6. Conclusions

FMD studies on humans have provided encouraging but still conflicting results on its effect in improving metabolic health. These studies suggest that the higher the BMI and the severity of metabolic disorders, the greater the benefits provided by FMD. Additional studies with longer duration and larger participants’ numbers should add more important information for understanding FMD metabolic effects. Also, targeting different age groups and comparing the effects in relation to gender should also provide more knowledge in creating individualized nutritional recommendations.

## Figures and Tables

**Table 1 metabolites-15-00150-t001:** Overview of Clinical Studies on FMD Interventions and Their Impact on Weight and BMI.

Author	Study Design	Intervention	Population	Age (Years Old)	Number	Length of Study	Weight (kg) and WC (cm) Changes	BMI (kg/m^2^)	Dropout Rate
Kim et al. [58]	Clinical Study	modified fasting therapy (MFT)	obese patients	39 ± 9	26 (17 women)	35 days	↓ 5.16 ± 0.95 kg during MFT	NR	13.30%
Mishra et al. [59]	RCT	FMD vs. Mediterranean Diet (MD)	MetS (obesity and hypertension)	54.86 ± 12.38	84 (FMD: *n* = 44; 32 women)	4 months (four cycles FMD-5 days/month) and 3 months follow-up	↓ BW (FMD: −7.8 ± 1.3 lbs vs. MD: −9.3 ± 1.2 lbs]; ↓ WC (FMD: −1.4 ± 0.4 inch vs. MD: −1.9 ± 0.3 inch)	NS	15.90%
Maifeld et al. [55]	RCT	FMD vs. DASH	MetS (obesity and hypertension)	58 ± 8	70 (44 women)	3 months (5 days of FMD followed by 3 months of DASH)	↓ BW in fasting + DASH group;	NR	NR
Nardon et al. [49]	RCT	FMD	healthy young men (BMI 18.5–30)	22.3 ± 1.2	24 (24 men)	90–100 days (three cycles FMD-5 days/month)	↓ 2.6 kg after the first cycle	NR	16.70%
Videja et al. [50]	Interventional Study	FMD vs. regular diet with added vegetables (VEG)	healthy, slightly overweight	39 ± 2	43 (FMD: *n* = 24; 9 men)	1 cycle FMD—5 days	↓ 2.8 ± 0.2 kg (FMD)	↓ 0.9 ± 0.06 units FMD vs. 0.09 ± 0.05 units VEG	2.30%
Brandhorst et al. [11]	Pilot Clinical Trial	FMD	generally healthy adults	27.6–70	38 (FMD: *n* = 19; 7 women)	3 months (three cycles of 5 days of FMD followed by 25 days of normal food intake)	↓ 3% in BW (3.1% ± 0.3%; *p* < 0.001)	NR	19.00%
Brandhorst et al. [60]	RCT	FMD	mixed (healthy and prediabetic adults)	43.3 ± 11.7	100 (FMD: *n* = 52; 33 men)	3 months (three cycles of 5 days of FMD); follow-up 3 months	NR	↓ (*p* = 0.0002)	25.00%
Sadeghian et al. [57]	RCT	FMD vs. CER (daily 500 kcal deficit)	metabolically healthy women with obesity	34.03 ± 1.29	60 women (FMD: *n* = 30)	2 months (two cycles of 5 days of FMD)	NS: mean weight change for CER − 2.29 (SD = 1.95) kg; FMD − 1.13 (SD = 2.27) kg (*p* = 0.06).	NR	20.00%
Wei et al. [48]	RCT	FMD vs. normal diet	generally healthy adults, mixed BMIs	43.3 ± 11.7	100 (FMD: *n* = 52, women = 33)	3 months (three cycles of 5 days of FMD)	FMD: ↓ 2.6 ± 2.5 kg (*p* < 0.001); Control: NS weight change	NR	25.00%
Sulaj et al. [56]	RCT	FMD vs. Mediterranean diet	T2DM with nephropathy (BMI 23–40 kg/m^2^)	64.9 ± 1.6	40 participants (FMD: *n* = 21, women = 6)	6 months (six cycles of 5 days of FMD) and 3 months follow-up	↓ −7.2 kg [−7.5% (−9.4, −4.9), *p* ≤ 0.001] in the FMD group, vs. −1.1kg [−1.2% (−2.9, 0.7)] in the M-Diet group.	NR	17.50%
Tang et al. [51]	RCT	FMD with specific meal replacements vs. standard meal replacements	T2DM; BMI ≥ 28 kg/m^2^	50.02 ± 1.76	100 (FMD: *n* = 50, women = 38	4 months (FMD meal replacement 5 days in the second week of a month and normal diet for the rest of the month)	↓ WC in the FMD group	↓ 25.04 ± 1.00 kg/m^2^ in FMD vs. 28.99 ± 0.99 in controls	14.00%
Van den Burg et al. [53]	RCT	FMD vs. usual care	T2DM and BMI ≥ 27 kg/m^2^	62 ± 8	92 (FMD: *n* = 49, women = 23).	12 months (a monthly 5-consecutive day FMD program)	↓ BW: −3.6 kg; 95% CI −5.2, −2.1; *p* < 0.001); ↓ WC (−3.5 cm; 95% CI −5.3, −1.6; *p* < 0.001)	↓ −1.2 kg/m^2^ (95% CI: −1.7–−0.7; *p* < 0.001),	20.00%
Van der Velden et al. [61]	RCT	FMD vs. Glycocalyx mimetic supplementation (Endocalyx)	South-Asian Surinamese patients with T2DM	61 ± 6	56 (FMD: *n* = 18, women = 12)	3 months (three cycles of 5 days of FMD) and follow-up at 6 months	NR	FMD: ↓ −1.0 kg/m^2^; Endocalyx: NS	30.00%
Kender et al. [52]	RCT	FMD vs. Mediterranean diet	T2DM; mixed	66.6 ± 5.8	31 (FMD: *n* = 17; women = 12)	6 months (three cycles of 5 days of FMD)	↓ FMD group 92.9 ± 3.5 kg vs. 85.8 ± 3.6 kg (*p* < 0.001)	↓ in FMD (30.1 ± 1.0 kg/m^2^ vs. 27.7 ± 1.0 kg/m^2^; *p* < 0.001)	NR

Legend: RCT—randomized control trial; MFT—modified fasting therapy; FMD—fasting-mimicking diet; CER—caloric energy restriction; VEG—regular diet with added vegetables; MD—Mediterranean diet; DASH—Dietary Approaches To Stop Hypertension; BMI—body mass index; BW-body weight; WC—weight circumference; NR—not reported; NS—not significant; SD—standard deviation; CI—confidence interval; ↓ decrease; ↑ increase.

**Table 2 metabolites-15-00150-t002:** Effects of FMD on fasting plasma glucose, HbA1c, and insulin resistance.

Author	Study Design	Population	Participants Number	Length of Study	FPG Levels (mg/dL)	HbA1c (%)	HOMA-IR
Mishra et al. [59]	RCT	MetS (BMI ≥ 28 kg/m^2^, Hypertension)	84 (FMD *n* = 44)	4 months (3 months follow-up)	↓ FPG (*p* = 0.0488)	↓ (*p* = 0.0059); follow-up: ↓ (*p* = 0.0116)	↓ (*p* = 0.0475); follow-up— ↓ (*p* = 0.0066)
Maifeld et al. [55]	RCT	MetS (obesity and hypertension)	70	3 months	NR	NR	↑ insulin sensitivity
Videja et al. [50]	Clinical Trial	healthy, overweight	43 (*n* = 24) FMD	5 days	↓ FPG (−0.57 ± 0.11 mmol/L)	NR	↓ −0.55 ± 0.08
Brandhorst et al. [11]	Pilot Trial	generally healthy adults	38 (FMD: *n* = 19)	3 months	↓ FPG 11.3% ± 2.3% (*p* < 0.001)	NR	NR
Brandhorst et al. [60]	RCT	mixed (healthy and prediabetic adults)	100 (FMD: *n* = 52)	3 months	NR	↓ from 5.8 ± 0.3 to 5.43 ± 0.404 (*p* = 0.032)	NR
Huang et al. [76]	RCT	normal weight to overweight	105	1 day	NS.	NR	NR
Sadeghian et al. [57]	RCT	metabolically healthy women with obesity	60	2 months	↓ FPG in the CER group (*p* < 0.001).	NR	NS
Wei et al. [48]	RCT	generally healthy adults, mixed BMIs	100	3 months	↓ FPG in at-risk individuals	NR	NR
Sulaj et al. [56]	RCT	T2DM with nephropathy	40 (FMD: *n* = 21)	6 months (with 3 months follow-up)	↓ FPG in FMD group.	NR	↓ −3.8 (−5.6–−2.0); *p* ≤ 0.05
Tang et al. [51]	RCT	T2DM	100	4 months	FPG (5.25 ± 0.23 mmol/L in FMD vs. 6.27 ± 0.37 mmol/L in controls);	↓	NR
Van den Burg et al. [53]	RCT	T2DM	92 (FMD: *n* = 49)	12 months	↓ (*p* < 0.001).	↓ (−3.2 mmol/mol)	NR
Van der Velden et al. [61]	RCT	T2DM	56 (FMD: *n* = 18),	3 months (follow-up at 6 months)	NR	↓ (−5.1 mmol/mol during intervention)	NR
Kender et al. [52]	RCT	T2DM	31 (FMD: *n* = 17)	6 months	did not change in both groups	↓ in FMD group, NS changes in M-Diet group; (*p* <0.001)	NR

Legend: RCT—randomized control trial; FMD—fasting-mimicking diet; CER—caloric energy restriction; M-diet—Mediterranean diet; BMI—body mass index; FPG—fasting plasma glucose; HbA1c—glycated hemoglobin; HOMA-IR—Homeostatic Model Assessment Insulin Resistance; NR—not reported; NS—not significant; ↓ decrease; ↑ increase.

**Table 3 metabolites-15-00150-t003:** Summary of FMD Interventions and lipid profile changes.

Author	Intervention	Population	Number	Length of Study	Total Cholesterol (mg/dL)	LDLc (mg/dL)	HDLc (mg/dL)	Triglycerides (mg/dL)
Kim et al. [58]	MFT using fermented medicinal herbs	obese patients	26	35 days	↓ from 181.5 ± 17.4 to 156.0 ± 12.7 mg/dL	NR	NR	↓ from 103.9 ± 22.8 to 90.5 ± 18.4 mg/dL
Mishra et al. [59]	FMD vs. continuous Mediterranean diet	BMI ≥ 28, hypertension	84 (FMD: *n* = 44)	4 months (3 months follow-up)	↓ (FMD: −10.4 ± 4.3 mg/mL vs. MD: −10.7 ± 5.0 pg/mL)	NR	NR	NR
Videja et al. [50]	5-day FMD vs. regular diet with added vegetables	healthy, slightly overweight	43 (FMD: *n* = 24)	5 days	NS	NS	NS	↓ of up to 15% similar effects
Brandhorst et al. [60]	FMD cycles for 3 months	mixed (healthy and prediabetic adults)	100 (FMD: *n* = 52)	3 months (3 months follow-up)	↓	↓	NS	↓
Wei et al. [48]	FMD for 3 months vs. normal diet	healthy adults, mixed	100	3 months	NS	NS	↑ (*p* = 0.03)	NS
Sulaj et al. [56]	FMD vs. Mediterranean diet	T2DM with nephropathy	40 (FMD: *n* = 21)	6 months (3 months follow-up)	NS	NS	NS	NS
Tang et al. [51]	FMD with specific meal replacements vs. standard meal replacements	T2DM	100	4 months	↓ (*p* ≤ 0.001)	↓ (*p* ≤ 0.001)	↑ (*p* ≤ 0.001)	↓ (*p* ≤ 0.001)
Van den Burg et al. [53]	12-month FMD cycles vs. usual care for type 2 diabetes	T2DM	92 (FMD: *n* = 49)	12 months	NS	NS	↑ (*p* < 0.001)	NS
Van der Velden et al. [61]	FMD vs. Glycocalyx mimetic supplementation (Endocalyx)	T2DM	56 (FMD: *n* = 18)	3 months, follow-up at 6 months	NS	NS	NS	NS
Kender et al. [52]	FMD vs. Mediterranean diet	T2DM	31 (FMD: *n* = 17)	6 months	NS	NS	NS	NS

Legend: RCT—randomized control trial; MFT—modified fasting therapy; FMD—fasting-mimicking diet; BMI—body mass index; NS—not significant; NR—not reported; ↓ any decrease; ↑ any increase.

**Table 4 metabolites-15-00150-t004:** Effects of FMD on inflammatory markers, blood pressure, and antihypertensive medication use.

Author	Study Design	Intervention	Population	No. Participants	Inflammatory Markers	BP (mmHg)	Antihypertensive Medication
Maifeld et al. [55]	RCT	FMD vs. DASH	MetS (obesity and hypertension)	70	↓ pro-inflammatory markers (Th17 cells, TNF-α)	↓ 24 h ambulatory SBP and mean arterial pressure (*p* < 0.05)	↓ intake of antihypertensive medication in 43% (FMD) vs. 17% (DASH)
Brandhorst et al. [60]	RCT	FMD cycles;	healthy and prediabetic adults	100 (FMD: *n* = 52)	↓ CRP (*p* < 0.0001) in MetS patients	in MetS patients ↓ SBP (*p* < 0.0001)	NR
Wei et al. [48]	RCT	FMD 3 months vs. normal diet	generally healthy adults, mixed BMIs	100	↓CRP (*p* = 0.27)	↓ SBP (*p* = 0.023); ↓ DBP (*p* = 0.053)	NR
Sulaj et al. [56]	RCT	FMD vs. M-diet	T2DM with nephropathy (BMI 23–40 kg/m^2^)	40 (FMD: *n* = 21)	↓ suPAR by ~156 pg/mL after 6 cycles in FMD	NR	↓ in 10% (FMD) vs. 5% (M-Diet)
Tang et al. [51]	RCT	FMD with specific meal replacements vs. standard meal replacements	T2DM; BMI ≥ 28 kg/m^2^	100	NR	↓ SBP and DBP (*p* < 0.05)	NR
Van den Burg et al. [53]	RCT	12-month FMD vs. usual care for T2DMs	T2DM (BMI ≥ 27 kg/m^2^)	92 (FMD: *n* = 49)	NR	NS (SBP *p* = 0.87; DBP *p* = 0.74)	63% FMD and 79% control—similar anti-hypertensive medication.

Legend: RCT—randomized control trial; FMD—fasting-mimicking diet; DASH—Dietary Approach to Stop Hypertension; TNF—tumor necrosis factor; CRP—C reactive protein; sUPAR—soluble urokinase-type plasminogen activator receptor; SBP—systolic blood pressure; DBP—diastolic blood pressure; M-diet—Mediterranean diet; BMI—body mass index; ↓ decrease; ↑ increase.

## Data Availability

No new data was created or analyzed in this study. Data sharing is not applicable to this article.

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
