# Peer review of "Fasting Mimicking Diet for Metabolic Syndrome: A Narrative Review of Human Studies"

_metabolites, 2025, doi:10.3390/metabo15030150_

Round 1
Reviewer 1 Report
Comments and Suggestions for Authors
This is a very interesting and well documented narrative review on the effects of FMD in patients with MetS. FMD is a type of fasting diet that is not very broadly used and reviewed in comparison with TRF or 5:2. It is very helpful though that the review refers to all the studies on this type of fasting indicating the positive effects of this particular type of diet. The review has a very good structure, it covers all the aspects of MetS (obesity, diabetes, lipids, hypertenstion, MASLD) and it contains very useful and informative tables with overview of Clinical Studies on FMD Interventions and Their Impact on the components of MetS. As there are numerous studies and reviews on all the other types of fasting diets it is also necessary to present the effects of FMD in order to decide the use of this diet in the treatment of MetS as a significant proportion of the articles were dedicated to nutrition and metabolism. It is well mentioned that there are limitations such as high dropout rates, small sample sizes, and methodological constraints restrict the generalizability of current findings. I think that it a review that can be published and that it will be very useful for the scientific support of FMD.
Author Response
We are very grateful for your appreciation. We thank you for taking the time to review our paper.
Reviewer 2 Report
Comments and Suggestions for Authors
General comment:
The authors just focused on health status and did not discuss age and sex as effective factors in FMD. Are these factors effective in the Metabolic syndrome? If yes should be briefly discussed.
Abstract:
It is better that authors do not use “we” in the abstract.
Introduction:
Too long with unnecessary information. It is like a discussion. The authors should make it brief.
169-178: You explained too much about the obesity
179-198: You did not indicate how many days FMD lasted for each study.
Table 1: Did you have any analysis of the results of this table? How did you choose these articles? You should indicate the criteria for choosing the articles and studies.
Line 246: Please do not start the sentence with “this”
Line 302: What does the HOMA-IR stand for? The full type of words should be mentioned in the first use
Table 2: all abbreviated letters should be defined under the table. How did you choose these articles? You should indicate the criteria for choosing the articles and studies.
339: two parentheses should be merged.
359: What about other critical factors?
Table 3: Please define arrows under the table. How did you choose these articles? You should indicate the criteria for choosing the articles and studies.
Table: the initial letters of the main words are capitalized and this is not true in the title.
486: What does the NAFLD stand for? The full type of words should be mentioned in the first use
504-507: you explained briefly about the tumor necrosis factor-α without any indicated goal.
509-534: The authors should not discuss or add references in the conclusion. The conclusion should be written based on the most important findings and should have a strong message for future studies. The duration of FMD and the composition of the diet should be more focused on the conclusion.
Author Response
General comment:
- The authors just focused on health status and did not discuss age and sex as effective factors in FMD. Are these factors effective in the Metabolic syndrome? If yes should be briefly discussed.
Thank you, this is a very important comment. However, we did not identify a specific analysis of the metabolic effects of FMD related to gender. We discussed this aspect in the conclusions as follows:
Certain components of the metabolic syndrome (MetS) present with gender-specific differences in prevalence, magnitude, and physiological mechanisms. These are determined by hormonal, genetic, and behavioral factors, and manifest differently in cardiovascular and metabolic risk in men and women. Men tend to accumulate more visceral fat, have higher triglyceride levels, and lower HDL-C from a young age, leading to earlier cardiovascular events. In women, menopause marks a significant metabolic transition, with greater accumulation of visceral fat, insulin resistance, and hypertension. While LDL-C levels are a better predictor of cardiovascular disease in men, in women, risk is more closely correlated with triglycerides and inflammatory markers. Another important limitation is the lack of comparisons of the efficiency of FMD according to gender, although these are important differences in the prevalence of metabolic syndrome components that can be differently addressed by FMD.
- Abstract: It is better that authors do not use “we” in the abstract.
Thank you for pointing this out. We modified the sentence that included “we” as follows:
“In this narrative review, the effects of FMD in patients with MetS were analyzed, focusing on its impact on key metabolic components and summarizing findings from human studies”.
- Introduction: Too long with unnecessary information. It is like a discussion. The authors should make it brief.
Thank you for this remark. We shortened paragraphs included between lines 49-61 and we moved the paragraph between lines 69-84 in section 2.
- 169-178: You explained too much about the obesity
Thank you for pointing this out. We modified this paragraph as follows:
The diagnosis and evaluation of obesity have changed in recent periods. It is considered that the body mass index (BMI), initially used as diagnosis criteria, is not enough indicative for its echo on general health. WC is a better estimator of metabolic effects, being closely related to IR and thus with its comorbidities, such as MAFLD, dyslipidemia, or T2DM. According to Canadian Adult Obesity Clinical Practice Guidelines, in 2020 obesity was considered „a complex chronic disease in which abnormal or excess body fat (adiposity) impairs health, increases the risk of long-term medical complications and reduces lifespan” [46].
- 179-198: You did not indicate how many days FMD lasted for each study.
Thank you for this comment. Indeed, we did not include this data in the text, but they are presented in Table 1.
- Table 1: Did you have any analysis of the results of this table? How did you choose these articles? You should indicate the criteria for choosing the articles and studies.
This is a very important comment. We did not present the search strategy initially in the previous draft because we conducted a narrative review. To provide a more accurate description on the selection strategy of the studies included in our review we added a new section – Material and Methods and the following paragraph:
We performed a systematic search in two databases, PubMed and EBSCO using the term “fasting-mimicking diet” and a combination of keywords: “metabolic syndrome”, “obesity”, “hyperglycemia”, “insulin resistance”, ”dyslipidemia”, “high blood pressure”, “gut microbiota”, “metabolic associated fat liver disease”, “diabetes”. Using this search strategy we identified 193 studies. After excluding the duplicates (n=47), screening for titles and abstracts (n=72), and full texts (n=58), we selected 16 studies addressing FMD in humans with MetS or its components. We opted for a narrative review due to the heterogeneity of the studies’ design, length, and number of participants
- Line 246: Please do not start the sentence with “this”
We modified accordingly:
The authors' explanation was based on the long-lasting effects of the low protein/amino acid content of the FMD
- Line 302: What does the HOMA-IR stand for? The full type of words should be mentioned in the first use
Thank you for this observation. We added the full name for HOMA-IR in the text where it appeared for the first time.
- Table 2: all abbreviated letters should be defined under the table. How did you choose these articles? You should indicate the criteria for choosing the articles and studies.
Thank you for this useful comment, we modified it accordingly. Each table summarizes only the studies (from all the 16 identified studies) related to the parameters discussed in each paragraph and this was mentioned in the title of the table.
- 339: two parentheses should be merged.
Thank you for pointing this out. We modified it accordingly.
- 359: What about other critical factors?
Thank you for this comment, we chose to eliminate this discussion related to potential individual responses to FMD.
- Table 3: Please define arrows under the table. How did you choose these articles? You should indicate the criteria for choosing the articles and studies.
We modified the tables accordingly and we explained the selection of the studies.
- Table: the initial letters of the main words are capitalized and this is not true in the title.
We modified the tables according to your suggestion.
- 486: What does the NAFLD stand for? The full type of words should be mentioned in the first use
We replaced NAFLD with MAFLD in the line 486.
- 504-507: you explained briefly about the tumor necrosis factor-α without any indicated goal.
Thank you for this remark. We added additional information to be more explicit:
Liver macrophages, including Kupffer cells, monocyte-derived macrophages, and capsular macrophages, can undergo phenotypic changes in response to cytokines, fatty acids, endotoxins, and metabolites. These cells may adopt either a pro-inflammatory state (resembling classically activated M1-like macrophages) or an anti-inflammatory state (similar to alternatively activated M2-like macrophages). In the context of fatty liver disease, palmitic acid, a saturated fatty acid, promotes the polarization of macrophages towards the proinflammatory M1-like phenotype, which is reflected in increased production of TNF-α and IL-6
- 509-534: The authors should not discuss or add references in the conclusion. The conclusion should be written based on the most important findings and should have a strong message for future studies. The duration of FMD and the composition of the diet should be more focused on the conclusion.
Thank you for this suggestion. We added a discussion section, and we also focused on the effects of FMD duration and composition:
The number of cycles of fasting reported in the studies on the effect of FMD on MetS and its components has shown a broad variation, from 1 [50] to 12 cycles [56]. This variability limits the generality and potentially leads to heterogeneity of the re-sults on weight, anthropometric indices, and metabolic markers. Glycemia or lipid profile can respond immediately after changing the diet composition, but the dietary effects of FMD on weight, HbA1c, and even inflammation markers need a longer peri-od to be observed. Longitudinal studies with longer lengths are necessary to better un-derstand the effect of FMD on the metabolic markers associated with MetS.
FMD is a short-term dietary intervention, which provides a low protein intake, with a high amount of healthy fats. At the same time, this diet brings the necessary complex carbohydrates, as well as essential vitamins and minerals to avoid potential side effects determined by essential nutrient deficits [66]. The low protein content of FMD aims to decrease the IGF-1 signaling pathway, which is an essential mechanism to promote longevity and cells resistance to stress. Previous studies have shown that high levels of IGF-1 are associated with an accelerated rate of aging and high risk of cancer and metabolic diseases. The downregulation of the IGF-1/mTOR pathway en-hances autophagy and cell protection mechanisms [67]. Diets with a low content of proteins and specific amino acids, such as methionine, decrease IGF-1 signaling pathways and enhance mitochondrial function, leading to lifespan extension in animal models [68]. The high intake of unsaturated lipids provided by FMD is essential for ketogenesis triggering. Thus, the production of β-hydroxybutyrate and acetoacetate al-lows their use as an alternative fuel to glucose, mimicking the metabolic effects of pro-longed fasting [69]. Not only that the ketone bodies provide energy during caloric re-striction, but at the same time they exert neuroprotective and anti-inflammatory ef-fects and also regulate gene expression by inhibiting histone deacetylases [70]. While strict fasting diets have a severe deficit of carbohydrates, FMD includes a controlled quantity of complex carbohydrates, which ensures a minimal glucose intake, necessary to maintain energy homeostasis without inducing a significant insulin rise [11]. Furthermore, this low level of carbohydrates helps reduce systemic inflammation and enhance insulin sensitivity, which are both important mechanisms in T2DM and MetS prevention [53].
Moreover, we introduced a new conclusion section to emphasize the most important findings and further messages for future studies.
FMD studies on humans have provided encouraging but still conflicting results on its effect in improving metabolic health. These studies suggest that the higher the BMI and the severity of metabolic disorders, the greater the benefits provided by FMD. Additional studies with longer duration and larger participants’ numbers should add more important information for understanding FMD metabolic effects. Also targeting different age groups and comparison of the effects in relation to gender should also provide more knowledge in creating individualized nutritional recommendations.
Reviewer 3 Report
Comments and Suggestions for Authors
1. It is necessary to include the participants' ages in the tables. If the study involves different age groups such as children, adolescents, adults, and those aged 65+, the differences between these age categories should be discussed in the paper.
2. Ensure that the text includes details about the search criteria and the databases used.
3. Line 62-65 - You have cited the same reference three times consecutively and relied on a single article to support the entire paragraph. To improve clarity and scientific rigor, consolidate the citations where appropriate and incorporate additional relevant sources to strengthen the argument.
The same issue appears in other parts of the text, for example, with reference 12, 46 etc. Please revise accordingly.
Line 188 add SD or () value , as you use this format to present results in the next sentence.
Line 360 - replace "mixed results" with something more appropriate for academic paper (for example -inconsistent findings)
Line 373 delete "and"
Some sentences have to be improve to get a more formal, scientific tone. For example: line 408 - the effects may be negligible - check whole paper.
4. It is common in some fields, particularly in early-stage research or exploratory studies, to interpret p-values between 0.05 and 0.1 as suggestive of a trend. Therefore, I suggest adding that part where you explain that p = 0.06, but it is not significant.
5. The section 3.2 discusses both human and animal models, but it is not clearly indicated which study was conducted on which model. Since this review focuses on human models, I kindly ask that the division between the two be made more explicit, so that it is immediately clear which study was performed on which model.
6. The text from lines 514-518 is more suitable for the discussion section than for the conclusion.
Author Response
- It is necessary to include the participants' ages in the tables. If the study involves different age groups such as children, adolescents, adults, and those aged 65+, the differences between these age categories should be discussed in the paper.
Thank you for your important suggestion. However, we did not identify a specific analysis of the metabolic effects of FMD related to gender. We discussed this aspect in the conclusions as follows:
Certain components of the metabolic syndrome (MetS) present with gender-specific differences in prevalence, magnitude, and physiological mechanisms. These are determined by hormonal, genetic, and behavioral factors, and manifest differently in cardiovascular and metabolic risk in men and women. Men tend to accumulate more visceral fat, have higher triglyceride levels, and lower HDL-C from a young age, leading to earlier cardiovascular events. In women, menopause marks a significant metabolic transition, with greater accumulation of visceral fat, insulin resistance, and hypertension. While LDL-C levels are a better predictor of cardiovascular disease in men, in women, risk is more closely correlated with triglycerides and inflammatory markers. Another important limitation is the lack of comparisons of the efficiency of FMD according to gender, although these are important differences in the prevalence of metabolic syndrome components that can be differently addressed by FMD.
- Ensure that the text includes details about the search criteria and the databases used.
This is a very important comment. We did not present the search strategy initially in the previous draft because we conducted a narrative review. To provide a more accurate description on the selection strategy of the studies included in our review we added a new section – Material and Methods and the following paragraph:
We performed a systematic search in two databases, PubMed and EBSCO using the term “fasting-mimicking diet” and a combination of keywords: “metabolic syndrome”, “obesity”, “hyperglycemia”, “insulin resistance”, ”dyslipidemia”, “high blood pressure”, “gut microbiota”, “metabolic associated fat liver disease”, “diabetes”. Using this search strategy we identified 193 studies. After excluding the duplicates (n=47), screening for titles and abstracts (n=72), and full texts (n=58), we selected 16 studies addressing FMD in humans with MetS or its components. We opted for a narrative review due to the heterogeneity of the studies’ design, length, and number of participants
- Line 62-65 - You have cited the same reference three times consecutively and relied on a single article to support the entire paragraph. To improve clarity and scientific rigor, consolidate the citations where appropriate and incorporate additional relevant sources to strengthen the argument.
This is a very important point. We chose this reference because the aim of this paragraph was to present the diet characteristics as described by the author of the FMD. (Brandhorst S, Choi IY, Wei M, Cheng CW, Sedrakyan S, Navarrete G, Dubeau L, Yap LP, Park R, Vinciguerra M, Di Biase S, Mirzaei H, Mirisola MG, Childress P, Ji L, Groshen S, Penna F, Odetti P, Perin L, Conti PS, Ikeno Y, Kennedy BK, Cohen P, Morgan TE, Dorff TB, Longo VD. A Periodic Diet that Mimics Fasting Promotes Multi-System Regeneration, Enhanced Cognitive Performance, and Healthspan. Cell Metab. 2015 Jul 7;22(1):86-99. doi: 10.1016/j.cmet.2015.05.012. Epub 2015 Jun 18. PMID: 26094889; PMCID: PMC4509734.). We removed the reference number after each sentence and kept it at the end of the paragraph.
- The same issue appears in other parts of the text, for example, with reference 12, 46 etc. Please revise accordingly.
Thank you for pointing this out. The paragraph citing reference 12 provided details on the results of a recent bibliometric analysis. We could not find other reviews related to the extension of the research on FMD during the last years. The paragraphs containing reference 46 represents the recommendations of the Canadian Adult Obesity Clinical Practice Guidelines. We also add citation after the following sentence - WC is a better estimator of metabolic effects, being closely related to IR and thus with its comorbidities, such as MAFLD, dyslipidemia, or T2DM [46]
- Line 188 add SD or () value , as you use this format to present results in the next sentence.
Thank you for pointing this out. We changed the format according to yout recommensation:
WC reduced significantly in the FMD group (from 105.10 ± 5.20 cm to 90.82 ± 4.26 cm) compared to the control group (from 104.32 ± 4.84 cm to 98.38 ± 4.27 cm)
- Line 360 - replace "mixed results" with something more appropriate for academic paper (for example -inconsistent findings)
Thank you for pointing this out. We changed this as follows:
Some studies have shown inconsistent findings, with improvements in some metabolic parameters but not others
- Line 373 delete "and"
We modified the sentence as follows:
Higher plasma levels of free fatty acids correlate with the overproduction of VLDL.
- Some sentences have to be improve to get a more formal, scientific tone. For example: line 408 - the effects may be negligible - check whole paper.
Thank you for pointing this out. We changed this sentence:
Not all patients responded favorably, and in some cases (patients with T2DM with nephropathy [56]), the effects may be negligible.
- It is common in some fields, particularly in early-stage research or exploratory studies, to interpret p-values between 0.05 and 0.1 as suggestive of a trend. Therefore, I suggest adding that part where you explain that p = 0.06, but it is not significant.
Thank you for pointing this out. We modified the sentence to emphasize the negative trend related to weight change as follows:
Although Sadeghian et al. did not report a significant weight change (CER − 2.29 ([SD], 1.95) kg vs. − 1.13 (2.27) kg for FMD (p = 0.06), that may indicate a negative trend.
- The section 3.2 discusses both human and animal models, but it is not clearly indicated which study was conducted on which model. Since this review focuses on human models, I kindly ask that the division between the two be made more explicit, so that it is immediately clear which study was performed on which model.
Thank you for pointing this out. We modified the paragraph in order to clearly differentiate between the results provided by animal studies and those from human studies:
FMD may benefit patients with chronic metabolic-related diseases such as T2DM and MAFLD [18]. Cheng et al. [69] found that a 4-day FMD in mice has been shown to stimulate the regeneration of β cells, suggesting its potential contribution to the improvement in both type 1 and type 2 diabetes. The author also performed a study on a small sample of human subjects and on human pancreatic β-cells, which highlight that the inhibition of mTOR, along with PKA, promotes Ngn3-driven β-cell regeneration in human type 1 diabetes islets [69]. Under fasting conditions PKA and mTOR activity in human pancreatic islets with type 1 diabetes decreases, stimulating the expression of specific genes and subsequent insulin production [70]. mTORC1 plays an essential role in the survival and function of human pancreatic β cells by modulating autophagy and reducing endoplasmic reticulum stress [71]. Fasting influences mTORC1 and mTORC2, affecting glucagon secretion and energy homeostasis in mice with tissue-specific deletion of the mTORC1 regulator Raptor in α cells (αRaptorKO) [72]. Multiple other studies performed on animal models underscored the potential of fasting in influencing endocrine pancreatic function. The inhibition of mTOR promotes autophagy by recycling cellular components and reducing oxidative stress, facilitating the adaptation of pancreatic β-cells to the conditions of reduced energetic intake [73]. This process contributes to maintaining optimal insulin secretion and protecting against endoplasmic reticulum stress-induced apoptosis, highlighting the essential role of mTOR in regulating energy homeostasis and preventing the progression of metabolic dysfunction in diabetes [74]. The interaction between the mTOR signaling pathway and the PKA pathway is achieved through the phosphorylation of the Raptor protein, a key component of the mTORC1 complex [75]. This phosphorylation is essential for glucose metabolism, with mice with a phosphorylation-resistant Raptor mutation exhibiting metabolic dysfunction, including IR [75]. On the other hand, PKA can inhibit mTORC1 activity through the same phosphorylation site, thus suggesting a dual regulation depending on the cellular context [76]. The mTORC1 complex acts as a metabolic rheostat, integrating signals from nutrients, energy, and growth factors to regulate processes such as protein synthesis and autophagy, thereby influencing cell growth and metabolism [77].
- The text from lines 514-518 is more suitable for the discussion section than for the conclusion.
Thank you for this suggestion. We added a discussion section, and we also focused on the effects of FMD duration and composition:
The number of cycles of fasting reported in the studies on the effect of FMD on MetS and its components has shown a broad variation, from 1 [50] to 12 cycles [56]. This variability limits the generality and potentially leads to heterogeneity of the re-sults on weight, anthropometric indices, and metabolic markers. Glycemia or lipid profile can respond immediately after changing the diet composition, but the dietary effects of FMD on weight, HbA1c, and even inflammation markers need a longer peri-od to be observed. Longitudinal studies with longer lengths are necessary to better un-derstand the effect of FMD on the metabolic markers associated with MetS.
FMD is a short-term dietary intervention, which provides a low protein intake, with a high amount of healthy fats. At the same time, this diet brings the necessary complex carbohydrates, as well as essential vitamins and minerals to avoid potential side effects determined by essential nutrient deficits [66]. The low protein content of FMD aims to decrease the IGF-1 signaling pathway, which is an essential mechanism to promote longevity and cells resistance to stress. Previous studies have shown that high levels of IGF-1 are associated with an accelerated rate of aging and high risk of cancer and metabolic diseases. The downregulation of the IGF-1/mTOR pathway en-hances autophagy and cell protection mechanisms [67]. Diets with a low content of proteins and specific amino acids, such as methionine, decrease IGF-1 signaling pathways and enhance mitochondrial function, leading to lifespan extension in animal models [68]. The high intake of unsaturated lipids provided by FMD is essential for ketogenesis triggering. Thus, the production of β-hydroxybutyrate and acetoacetate al-lows their use as an alternative fuel to glucose, mimicking the metabolic effects of pro-longed fasting [69]. Not only that the ketone bodies provide energy during caloric re-striction, but at the same time they exert neuroprotective and anti-inflammatory ef-fects and also regulate gene expression by inhibiting histone deacetylases [70]. While strict fasting diets have a severe deficit of carbohydrates, FMD includes a controlled quantity of complex carbohydrates, which ensures a minimal glucose intake, necessary to maintain energy homeostasis without inducing a significant insulin rise [11]. Furthermore, this low level of carbohydrates helps reduce systemic inflammation and enhance insulin sensitivity, which are both important mechanisms in T2DM and MetS prevention [53].
Moreover, we introduced a new conclusion section to emphasize the most important findings and further messages for future studies.
FMD studies on humans have provided encouraging but still conflicting results on its effect in improving metabolic health. These studies suggest that the higher the BMI and the severity of metabolic disorders, the greater the benefits provided by FMD. Additional studies with longer duration and larger participants’ numbers should add more important information for understanding FMD metabolic effects. Also targeting different age groups and comparison of the effects in relation to gender should also provide more knowledge in creating individualized nutritional recommendations.
Reviewer 4 Report
Comments and Suggestions for Authors
Thank you very much for the opportunity to review the article „Fasting Mimicking Diet for Metabolic Syndrome: a Narrative Review of Human studies”. The article is generally well written and covers an important and interesting topic.
Below are my comments and suggestions:
- The article should provide details on how articles were selected for the literature review (what databases were searched, what keywords and operators were used, what criteria for including and excluding articles were used)
- In some cases, footnotes in the text were written incorrectly, for example table 1, table 2 (A. Mishra et al. [62]) – the initial of the name should not be written
- Inconsistent font for table titles – sometimes bold, sometimes not
- Table 1 – in the BMI column, please write down the unit
- Table 1 – Dropout Rate – in my opinion the same rounding should be used everywhere
- Table 1 – all abbreviations and symbols used in the table should be explained below the table
- Table 2 - all abbreviations and symbols used in the table should be explained below the table
- Table 3 - all abbreviations and symbols used in the table should be explained below the table
- In some cases, units are missing in table column titles - please verify all tables
Author Response
Thank you very much for the opportunity to review the article „Fasting Mimicking Diet for Metabolic Syndrome: a Narrative Review of Human studies”. The article is generally well written and covers an important and interesting topic.
Thank you for your kind appreciation.
Below are my comments and suggestions:
The article should provide details on how articles were selected for the literature review (what databases were searched, what keywords and operators were used, what criteria for including and excluding articles were used)
This is a very important comment. We did not present the search strategy initially in the previous draft because we conducted a narrative review. To provide a more accurate description on the selection strategy of the studies included in our review we added a new section – Material and Methods and the following paragraph:
We performed a systematic search in two databases, PubMed and EBSCO using the term “fasting-mimicking diet” and a combination of keywords: “metabolic syndrome”, “obesity”, “hyperglycemia”, “insulin resistance”, ”dyslipidemia”, “high blood pressure”, “gut microbiota”, “metabolic associated fat liver disease”, “diabetes”. Using this search strategy we identified 193 studies. After excluding the duplicates (n=47), screening for titles and abstracts (n=72), and full texts (n=58), we selected 16 studies addressing FMD in humans with MetS or its components. We opted for a narrative review due to the heterogeneity of the studies’ design, length, and number of participantsIn some cases, footnotes in the text were written incorrectly, for example table 1, table 2 (A. Mishra et al. [62]) – the initial of the name should not be written
Inconsistent font for table titles – sometimes bold, sometimes not
Table 1 – in the BMI column, please write down the unit
Table 1 – Dropout Rate – in my opinion the same rounding should be used everywhere
Table 1 – all abbreviations and symbols used in the table should be explained below the table
Table 2 - all abbreviations and symbols used in the table should be explained below the table
Table 3 - all abbreviations and symbols used in the table should be explained below the table
In some cases, units are missing in table column titles - please verify all tables
Thank you for your observation. We modified the content of table according to your recommendations.
Round 2
Reviewer 2 Report
Comments and Suggestions for Authors
Thank you for the revision
Reviewer 3 Report
Comments and Suggestions for Authors
authors addressed all comments